

# Effect of ethyl methanesulfonate mediated mutation for enhancing morpho-physio-biochemical and yield contributing traits of fragrant rice

Areeqa Shamshad[1], Muhammad Rashid[1], Ljupcho Jankuloski[2], Kamran Ashraf[3,4], Khawar Sultan[5], Saud Alamri[6], Manzer H. Siddiqui[6], Tehzeem Munir[5] and Qamar uz Zaman[5]

[1] Nuclear Institute for Agriculture and Biology College (NIAB-C), PIEAS, Islamabad, Pakistan
[2] International Atomic Energy Agency, Joint FAO/IAEA Centre, Plant Breeding and Genetics Section, Vienna, Austria
[3] Department of Bioengineering and Biotechnology, School of Biotechnology, Kunming University of Science and Technology, Shanghai, China
[4] Department of Food Sciences, Government College University Faisalabad, Sahiwal Campus, Faisalabad, Pakistan
[5] Department of Environmental Sciences, The University of Lahore, Lahore, Pakistan
[6] Department of Botany and Microbiology, College of Science, King Saud University, Riyadh, Saudi Arabia

Corresponding author
Qamar uz Zaman,
qamar.zaman1@envs.uol.edu.pk

## ABSTRACT

**Background**. Chemical mutagenesis has been successfully used for increasing genetic diversity in crop plants. More than 800 novel mutant types of rice (*Oryza sativa* L.) have been developed through the successful application of numerous mutagenic agents. Among a wide variety of chemical mutagens, ethyl-methane-sulfonate (EMS) is the alkylating agent that is most commonly employed in crop plants because it frequently induces nucleotide substitutions as detected in numerous genomes.

**Methods**. In this study, seeds of the widely consumed Basmati rice variety (Super Basmati, *Oryza sativa* L.) were treated with EMS at concentrations of 0.25%, 0.50%, 0.75%, 1.0%, and 1.25% to broaden its narrow genetic base.

**Results**. Sensitivity to a chemical mutagen such as ethyl methanesulfonate (EMS) was determined in the M1 generation. Results in M1 generation revealed that as the levels of applied EMS increased, there was a significant reduction in the germination percent, root length, shoot length, plant height, productive tillers, panicle length, sterile spikelet, total spikelet, and fertility percent as compared to the control under field conditions. All the aforementioned parameters decreased but there was an increase in EMS mutagens in an approximately linear fashion. Furthermore, there was no germination at 1.25% of EMS treatment for seed germination. A 50% germination was recorded between 0.50% and 0.75% EMS treatments. After germination, the subsequent parameters, viz. root length and shoot length had $LD_{50}$ between 05.0% and 0.75% EMS dose levels. Significant variation was noticed in the photosynthetic and water related attributes of fragrant rice. The linear increase in the enzymatic attributes was noticed by the EMS mediated treatments. After the establishment of the plants in the M1 generation in the field, it was observed that $LD_{50}$ for fertility percentage was at EMS 1.0% level, for the rice variety.

**Conclusion**. Hence, it is concluded that for creating genetic variability in the rice variety (Super Basmati), EMS doses from 0.5% to 0.75% are the most efficient, and effective.

## INTRODUCTION

Nearly half of the world population relies mostly on the rice crop as a primary staple food. An agro-ecological landscape and the associated biodiversity and customer quality choices have also been major contributors to the development of new rice varieties which is leading to more genetic diversification and several varietal groupings (*Loko et al., 2021*). To ensure food security, it is essential to increase production by utilizing effective techniques for the efficient enhancement of yield (*Zaghum, Ali & Teng, 2022*). The aromatic local cuisine rice (*i.e.,* pulao or biryani) from the Indian subcontinent known as ''Basmati'' is made up of one such varietal group and is quite expensive both domestically and abroad. The extra-long, narrow grain, pleasant aroma, and fluffy soft textured Basmati rice variety its origins in the Himalayan foothills which provide the most suitable environment for its growth and its distinguishing features (*Hameed et al., 2019*; *Malabadi, Kolkar & Chalannavar, 2022*). Besides other breeding tools, irradiation (*i.e.,* fast neutrons, $\gamma$-rays, & X-rays) and chemical mutagens (*i.e.,* DEB, EMS, & sodium azide) have been frequently used to produce a broad range of functional mutations in rice (*Gulfishan et al., 2023*). Phenotypic characterization of Super Basmati using ethyl methanesulfonate (EMS) was carried out by earlier researchers (*Hameed et al., 2019*). For functional genomics and breeding studies, a large mutant population of coarse variety Katy was developed using EMS (*Jia et al., 2019*). In earlier studies, the upland rice variety Nagina 22, and the Japonica variety Shengdao 808 had also been used for developing mutants exhibiting tolerance to drought and salinity and natural variation studies (*Shang, Chun & Li, 2021*; *Zargar et al., 2022*).

Mutagens cause point mutations, making them suitable for creating missense and nonsense mutations that would result in functional mutations (*Shelar et al., 2021*). Ionizing radiation also causes chromosomal rearrangements and deletions (*Le Roux, 2019*; *Singh, Khar & Verma, 2021*). Mustard gas, methyl-methanesulfonate (MMS), EMS, and nitrosoguanidine are all alkylating agents that have diverse effects on DNA (*Ramesh et al., 2019*). According to *Talebi, Talebi & Shahrokhifar (2012)*, EMS produces mutations by alkylating guanine bases, which results in (mis)matches with thymine rather than cytosine and triggers transitions from G/C to A/T. EMS can also lead to A/T to G/C conversions through mismatches of 3-ethyladenine or G/C to T/A transversions by 7-ethylguanine hydrolysis (*Serrat et al., 2014*). The EMS causes point mutations in the rice genome and is one of the most commonly used mutagens in plants due to its potency and ease of application (*Upadhyaya et al., 2007*; *Husain, Bano & Khan, 2020*).

Since EMS induces an abundance of non-lethal point mutations (genome-wide), a slight mutant population (roughly ten thousand) is abundant to saturate the genome with

mutations (*Hernández-Muñoz et al., 2019*). The point mutation rate is four mutations per Mb in Arabidopsis (*Kazama et al., 2017*). A significant benefit of using a mutagen like EMS in forward genetic screens depends upon its efficacy in a range of organism types (*Taheri et al., 2017*). In different species, chemical mutagenesis induces a different rate of nucleotide substitutions. In Arabidopsis and maize mutational density was observed per gene (*Talebi, Talebi & Shahrokhifar, 2012*). EMS was also used in other crops like sugarcane where the calli were mutagenized with 0.5% EMS and exposed to 2% (w/v) PEG-6000 for induction of the osmotic stress (*Gadakh et al., 2021*). The mutant, dmc, was obtained from EMS treatment in wheat variety Guomai 301 (*Li et al., 2021*). New approaches have been undertaken in recent years to produce EMS-induced rice varieties at research institutes (*Kumawat et al., 2022*). The $LD_{50}$ dose is first calculated and then used to determine the best dose for inducing mutations in CR1009 and CR1009 sub1 rice (*Khannetah et al., 2021*). Thai highland rice (cv. Dawk Pa-yawm and Dawk Kha 50) was subjected to induced mutagenesis using EMS to create genetic variability (*Awais, Nualsri & Soonsuwon, 2019*; *Unan et al., 2022*). By leaving out this stage, the mutagen dose can result in either a high or low mutation frequency (*Barr & Fearns, 2016*; *Espina et al., 2018*; *Galal & Thabet, 2018*).

Basmati rice has a narrow genetic base and broadening its genetic base using non-Basmati rice material may affect its quality attributes. The transgenic approaches have GMO issues. Hence, changing genes in the living cell is not an easy job. The improvement of the CRISPR-Cas9 system in plants is carried out by gene editing which is more specific to gene removal or removing sequences. In CRISPR Cas9 the mutations are also random but often the intended changes are very precise. The removal of sequences/genes may have negative effects on disease resistance, drought, and salinity tolerance. The objective of CRISPR Cas9 is also to develop resistant plants against biotic and abiotic stresses. In the CRISPR editing system there is the

Possibility of altering off-target genes. However, the individuals resulting from the edit can be scrutinized to ensure that there are no additional changes. This can consume time and significant financial resources. It is possible that, in this sense, the use of EMS may be a better option.

In CRISPR Cas9, the bacterial system is used to protect from viruses and replace the mutant/lethal gene with the healthy copy. This can be done by adding the other DNA that carries the desired sequence in cultured cells. In the mutation breeding approach, mutants developed mostly due to deletions in DNA sequences, and no transgenic approach is involved. Like CRISPR Cas9, the cultured cells (Callus) are screened on hygromycin media to conform to the transgenic. Basmati rice being an exportable commodity, the hygromycin resistance is an issue in Basmati rice patenting. The induced mutation has the option to alter one or the other desirable genes without compromising other traits.

Besides, among the conventional breeding and transgenic approaches, induced mutation is the easiest approach that can be used for creating genetic variability without compromising the quality attributes. Inducing genetic diversity in rice by ionizing radiation has been proven to be successful. Understanding the relative biological effectiveness and efficiency of different mutagens is helpful in mutation breeding before the start of any sound breeding program. Many scientists have undertaken several experimental investigations in
this regard to identify the most efficient mutagenic method for the induction of desirable features in rice (*Jankowicz-Cieslak, Mba & Till, 2017*; *Rashid et al., 2009*).

This research aims at determining an optimal EMS dose (-50% lethal dose) in comparison to the standard (control) in order to generate variability, keeping typical characteristics of Super Basmati rice. Various EMS concentrations were applied to Super Basmati rice (Fragrant Rice) seeds and systematically assessed the survival and lethal doses during germination, seedling lethality, morpho-physio-biochemical, and yield attributes. Based on these observations, we may determine the optimal dose for EMS mutagenesis in the Super Basmati rice cultivar improvement against different ecological extremes conditions.

## MATERIALS & METHODS

### Plant material

In this research work a total of 400 seeds of the rice cultivar Super Basmati (*Oryza sativa* L. spp.) were selected by collecting from the Plant Breeding and Genetics Division of the Nuclear Institute for Agriculture and Biology (NIAB), Faisalabad, Pakistan.

### EMS mutagenesis

Super Basmati seeds were soaked in ultrapure water (-100 mL) up to a height of 5 cm above the seeds and stored at room temperature overnight for 20 h. The 50 mL of EMS with (v/v) concentrations (0.25%, 0.50%, 0.75%, 1.0%, and 1.25%) were added after decanting the ultrapure water (resistivity of 18.2 M Ω. cm at 25 °C). The seeds were then transferred and rinsed with 100 mL ultrapure water (five times and 4 min each) and 200 mL ultrapure water (four times, and 15 min each), and after that treated seeds were incubated at room temperature for 12 h. The processed seeds were then washed in continuously running tap water for about 4 h before being placed in Petri dishes for further analysis (*Talebi, Talebi & Shahrokhifar, 2012*) (Fig. 1).

### Photosynthetic attributes

Photosynthetic rate (*A*), transpiration rate (*E*), and stomatal conductance (*gs*) were measured on completely lengthened uppermost leaves with a portable photosynthesis system(Infra-Red Gas Analyzer) at a light saturating intensity between 9:00 am to 12:00 noon on a full-sun day. Samples (∼5 g) of leaves from each treatment were collected on test tubes containing acetone (85% each sample) were macerated in the same tube. Samples were kept in the dark for 24 h to allow the extraction of photosynthetic pigments. Tubes were centrifuged for 10 min at 4,000× at 4 °C to remove cellular debris. Supernatants were measured in a spectrophotometer (Halo DB-20/ DB-20S, UK) at 470, 647, and 664.5 nm to measure contents of chlorophyll (*Lichtenthaler, 1987*).

### Water related attributes

For the determination of water related attributes, three penultimate leaves of each treatment were harvested at the tillering stage. A pressure chamber was used to determine the leaf water potential. After that leaf samples were frozen and thawed sap was extracted and the osmotic potential was determined using digital Osmoter (Wescor, Logan, UT, USA) (*Farooq, Wahid & Lee, 2009*).

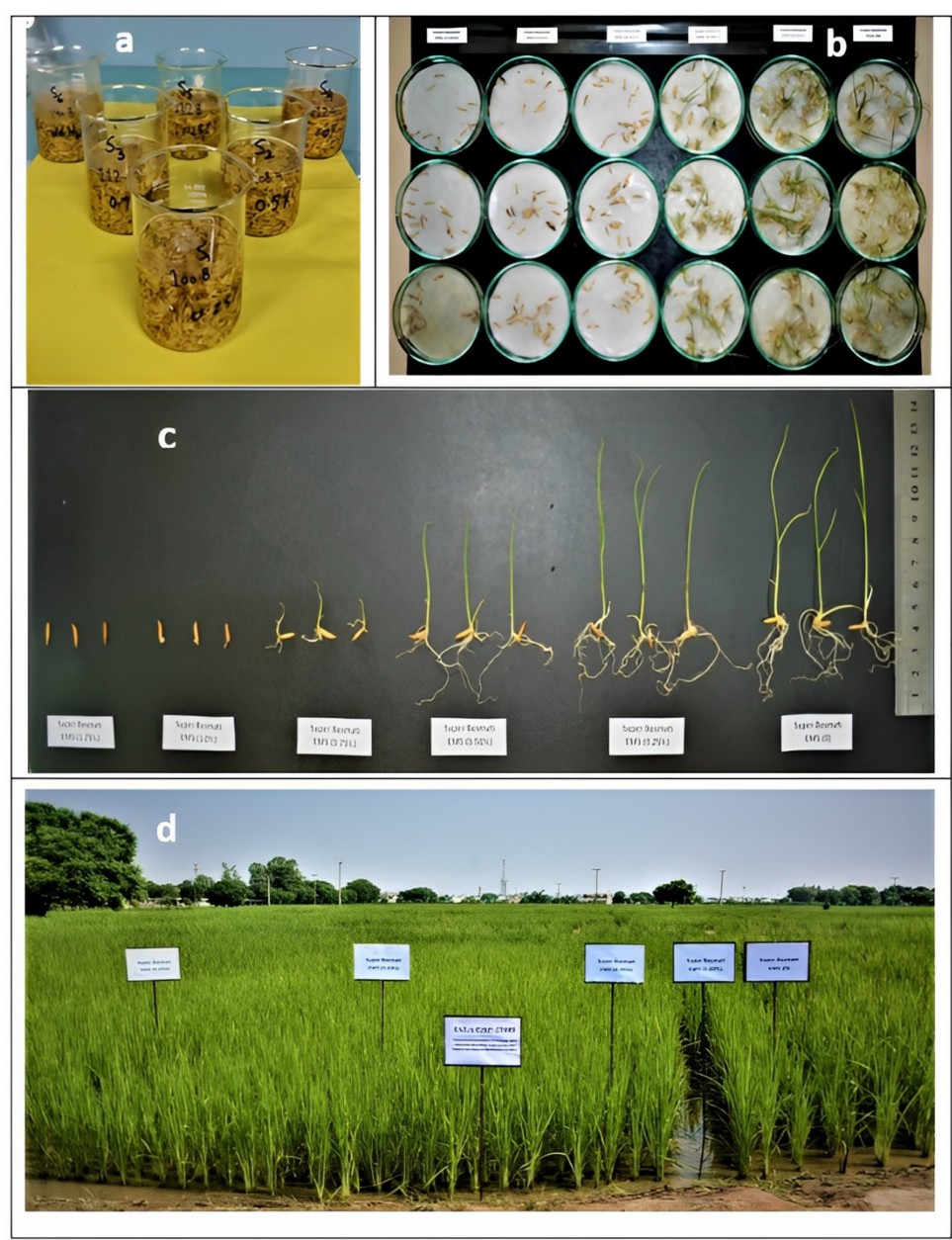

**Figure 1** Chemical induced mutagenesis in Super Basmati: (A) soaking paddy seeds of Super Basmati in 4 ultrapure water for 20 h, (B) sowing of EMS treated seeds of Super Basmati in Petri plates, (C) effect of 5 EMS on seedling, (D) sowing of EMS treated seed in the field.

## Activities of enzymatic antioxidants

An aliquot of fresh green leaf sample amount was homogenized with five mL of 50 mM Tris–HCl buffer (pH 8.0) for CAT and 50 mM $KH_2PO_4$ buffer (pH 7.0) for POX and APX determination. The homogenate was centrifuged at 5000 rpm for 20 min and the supernatant was then used as enzyme extract. The CAT (EC: 1.11.1.6) activity was assayed as

described by *Islam et al. (2009)*. The POX (EC: 1.11.1.7) and APX (EC: 1.11.1.11) activities were assayed as described by *Zeng, Liu & Xu (2011)*.

## Lethal dose study in EMS mutagenesis

In addition to the control, a total of about 40 seeds were planted on filter paper that had been dipped in 5 mL of ultrapure water in Petri plates following the EMS-induced treatments. Petri plates were then placed at 25 °C for 7 days in an incubator. The number of seeds that were grown under controlled conditions was counted and observed after seven days of germination. The germinated seeds from each applied EMS concentration, as compared to the control, were shifted to plastic pots and later in the rice field. In the greenhouse, the seedlings were irrigated with distilled water. After two weeks, the shoot and root lengths were measured using the sandwich blotter technique (*Ariraman et al., 2014*). After sown on the nursery bed, the emergence was recorded for each dose after germination. Parameters such as the height of the plant, panicle length, productive tillers, total spikelet, sterile spikelet, and fertility percent were measured at the physical maturity of rice plants.

## Statistical analysis

For lethal dose determination, the rice variety was treated with five levels of EMS concentrations and then sown in triplicate in a randomized block design. The least significant difference (LSD) test with $p$-values less than 0.05 was employed to analyze the average variance for all investigated parameters between treated and control plants. The statistical evaluation was carried out using Statistix 8.13 software. Principle component analysis (PCA) was carried out using Minitab-19.

## RESULTS

### Effect of EMS-induced mutagenesis on germination

The experimental data indicated that different doses of EMS caused variations in the germination of aromatic rice. The seed germination attributes of the control along with the treated seeds of Basmati rice are shown in Table 1. The major differences in the values of seed germination after the EMS treatment at varying doses of 0.25%, 0.50%, 0.75%, 1.00%, and 1.25% were highly significant at the 5% concentration. In all cases, there is an inhibitory effect, but it occurs to different degrees depending on the dose level. In the case of seed germination, data indicate that EMS had a retarding growth effect, or even inhibit it, depending on the dose applied, as compared to the control (Fig. 2). Under the conditions of EMS treatments of 0.25% and 0.50% levels, seeds showed the highest germination percentages, 91.4% & 89.6%, respectively, among all other EMS treatments. Low seed germination percentages were recorded in the higher doses at 0.75% and 1% levels to be 34.4% and 6.9%, respectively, while 1.25% treatment did not register any germination.

### Effect of EMS mutagenesis on seedling growth attributes

The length of the roots and shoots revealed that EMS-induced mutagenesis had a substantial effect on their growth as indicated by the size. According to the measurements and

| Table 1 | Mean value of germination attribute of fragrant rice by following EMS mutagenesis. | |
| --- | --- | --- |
| **Treatment** | **Germination** | |
| | **Observed** | **% Control** |
| Control | 19.33 | 100 |
| 0.25% | 17.67 | 91.4 |
| 0.50% | 17.33 | 89.6 |
| 0.75% | 6.66 | 34.4 |
| 1.00% | 1.33 | 6.9 |
| 1.25% | 0 | 0 |
| LSD% | 0.94 | |
| C.V% | 5.07 | |

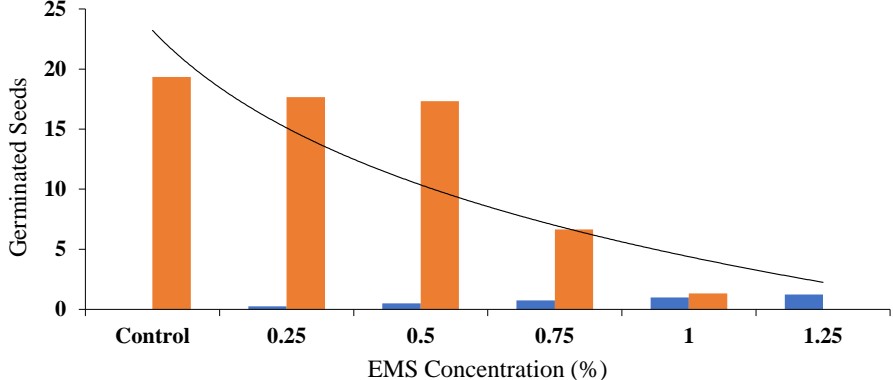

**Figure 2** Effect of different concentrations of EMS mutagenesis on seed germination of fragrant rice.

observations, shoot length decreased in proportion to the amount of EMS applied (Table 2 & Fig. 3). It was evident that when the concentration of EMS was increased, the root length decreased as compared to the control. The EMS 0.25% level exhibited the highest shoot length of 5.22 cm as compared to the control (5.44 cm). In other treatments of EMS viz. 0.5%, 1.0%, and 1.25%, a decreasing trend in shoot length of 4.85, 0.29, and 0.07 cm, respectively, were recorded as compared to the control. Among the EMS treatments, 0.25% level exhibited a maximum root length (5.75 cm) than the other treatments of 0.5% EMS (5.16 cm), 0.75% EMS (0.43 cm), and 1% EMS (0.15 cm). At a 1.0% concentration of EMS, the shortest root length was recorded in the experiment. A decreasing trend in the length of shoot and root was observed with the increase in the dose of EMS. When Basmati rice was treated with EMS concentrations greater than 1%, no seed germination was observed for the genotype under consideration.

## Effect of EMS mutagenesis on photosynthetic attributes

Various concentrations of EMS treatments significantly affected the photosynthetic attributes of fragrant rice. It was observed that by increasing the concentration of EMS treatments a linear increase was noticed up to 0.75% of EMS treatment. At 1%

**Table 2   Average value of root and shoot length attributes of fragrant rice by EMS-induced mutagenesis.**

| Treatment | Root length (cm) | | Shoot length (cm) | |
|---|---|---|---|---|
| | Observed | % Control | Observed | % Control |
| Control | 6.28 | 100 | 5.44 | 100 |
| 0.25% | 5.75 | 91.56 | 5.22 | 95.96 |
| 0.50% | 5.16 | 82.17 | 4.85 | 89.15 |
| 0.75% | 0.43 | 6.85 | 0.29 | 5.33 |
| 1.00% | 0.15 | 2.39 | 0.07 | 1.29 |
| 1.25% | 0 | 0 | 0 | 0 |
| LSD% | 0.16 | | 0.53 | |
| C.V% | 3.01 | | 11.34 | |

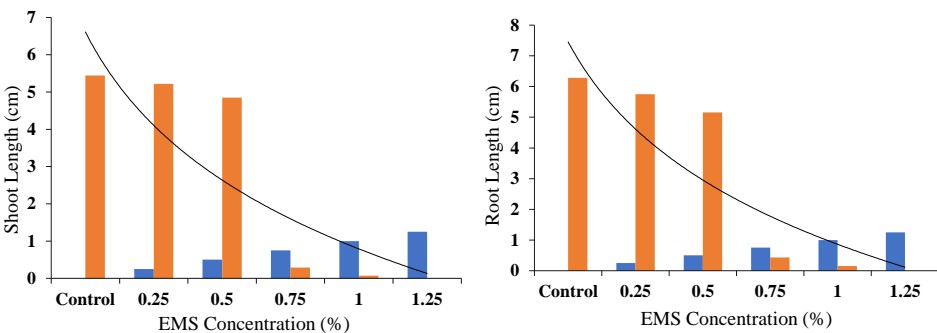

**Figure 3   Effect of different concentrations of EMS mutagenesis on root and shoot growth of fragrant rice.**

**Table 3   Effect of different concentrations of EMS mutagenesis on photosynthetic attributes of fragrant rice.**

| Treatments | Chlorophyll Contents (mg L$^{-1}$) | Photosynthetic Rate ($\mu$mol CO$_2$ m$^{-2}$ s$^{-1}$) | Transpiration Rate (mmol H$_2$O m$^{-2}$ s$^{-1}$) | Stomatal Conductance (mmol H$_2$O m$^{-2}$ s$^{-1}$) |
|---|---|---|---|---|
| Control | 4.87 B | 10.84 D | 4.83 E | 0.08 D |
| 0.25% | 5.03 B | 14.97 BC | 6.22 D | 0.06 CD |
| 0.50% | 5.30 A | 17.57 B | 9.04 B | 0.35 AB |
| 0.75% | 5.40 A | 13.96 A | 10.18 A | 0.41 A |
| 1.00% | 5.26 A | 12.33 CD | 8.33 C | 0.26 BC |
| 1.25% | 0.00 C | 0.00 E | 0.00 F | 0.00 D |
| HSD ($p \leq 0.05$) | 0.17 | 2.57 | 0.68 | 0.15 |

**Notes.**
Within each column, mean data followed by the same letters are not statistically different ($p \leq 0.05$ HSD test).

concentration, there was a decrease in all the photosynthetic attributes that was statistically at par with the 0.25% EMS treatment. The maximum of all the photosynthetic attributes was observed at 1.00% EMS treatment (Table 3).

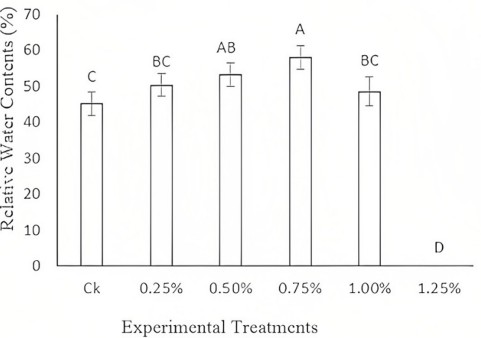
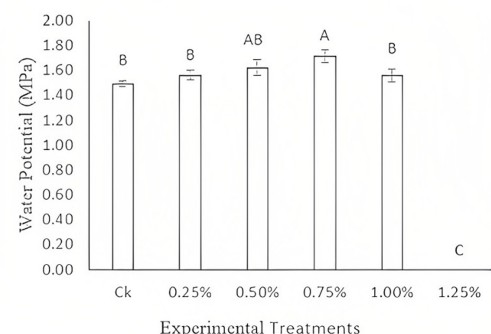
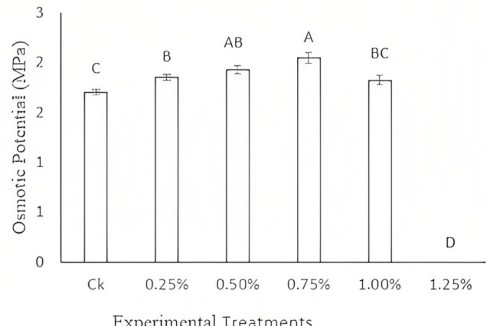

**Figure 4 Effect of different concentrations of EMS mutagenesis on water related attributes of fragrant rice.**

## Effect of EMS mutagenesis on water related attributes

It is observed from the experimental data of the current study that there exists a linear increase in the water related attributes after the EMS mutagenesis. Maximum relative water contents, water potential, and osmotic potential were noticed at 0.75% EMS treatment in the fragrant rice. However, at 1.00% there is a decrease in the water related attributes that was statistically at par with the concentration of 0.25%. A minimum of all the water related attributes was noticed under control conditions (Fig. 4).

## Effect of EMS mutagenesis on enzymatic antioxidants attributes

Significant variation was observed in the activities of enzymatic antioxidants by the EMS treatments. A linear increase was observed in all of the activities by the EMS treatments. A minimum activity was noticed at control while the maximum activity was observed at 1.00% EMS treatment. At 1.25% concentration, there was no value because of no germination (Table 4).

## Effect of EMS mutagenesis on phonological and yield contributing attributes

The maximum plant height (Table 5 & Fig. 5) after attaining maturity level was measured to be 104.6 cm and 105.0 cm in 0.25% EMS treatment and control, respectively. The minimum plant height (94.67 cm) was measured in 1% dose of EMS. In the present study,

**Table 4  Effect of different concentrations of EMS mutagenesis on enzymatic antioxidants of fragrant rice.**

| Treatments | APX activity ($\mu$mol/min/g FW) | POX activity ($\mu$mol/min/g FW) | CAT activity ($\mu$mol/min/g FW) |
|---|---|---|---|
| Control | 5.33 E | 9.10 E | 2.04 C |
| 0.25% | 8.65 D | 13.18 D | 2.86 C |
| 0.50% | 11.30 C | 16.30 C | 3.70 B |
| 0.75% | 14.30 B | 21.23 B | 4.11 B |
| 1.00% | 16.93 A | 27.97 A | 4.93 A |
| 1.25% | 0.00 F | 0.00 F | 0.00 D |
| HSD ($p \leq 0.05$) | 0.99 | 1.54 | 0.82 |

**Notes.**
Within each column, mean data followed by the same letters are not statistically different ($p \leq 0.05$ HSD test).

**Table 5  Mean value of phonological and yield attribute of fragrant rice by following EMS mutagenesis.**

| Treatment | Plant height (cm) | | Productive tillers | | Panicle length (cm) | |
|---|---|---|---|---|---|---|
| | Observed | % Control | Observed | % Control | Observed | % Control |
| Control | 104.97 | 100 | 3.90 | 100 | 26.9 | 100 |
| 0.25% | 104.57 | 99.62 | 3.67 | 94.10 | 26.9 | 100 |
| 0.50% | 104 | 99.08 | 3.60 | 92.31 | 25.63 | 95.28 |
| 0.75% | 103 | 98.12 | 3.50 | 89.74 | 25.3 | 94.05 |
| 1.00% | 94.67 | 90.19 | 3.17 | 81.28 | 24.67 | 91.71 |
| 1.25% | 0 | 0 | 0.00 | 0 | 0 | 0 |
| LSD% | 4.93 | | 0.52 | | 0.90 | |
| C.V% | 3.25 | | 9.87 | | 2.35 | |

EMS treatment at the highest concentration (1.25%) has shown an inhibitory effect as compared to the control. Productive tillers, panicle length, and total spikelet (Table 5) showed a maximum length of 3.67, 26.9, and 111.9 cm at 0.25% EMS treatments whereas control ranged as 3.90, 26.9, and 112.9 cm of productive tillers, panicle length, and total spikelet, respectively. However, at a 1.5% EMS level, an inhibitory effect for productive tillers, panicle length, and total spikelet was recorded as compared to the control. It was observed that the proportion of sterile spikelets increased as the applied EMS concentration increased. When Super Basmati was treated with a concentration of 0.25%, the highest value in sterile spikelets (85.2) was observed. Figure 3C indicates that as the applied EMS concentration was increased, fertility decreased. In the control group, the highest fertility rate (13.12%) was observed. When the Super Basmati cultivar was subjected to a concentration of 0.25%, the highest fertility rate (11.48%) was observed. At 1% EMS concentration, the lowest sterile spikelets, and fertility were registered. An EMS treatment above 1% concentration, showed a total inhibition of sterile spikelet, and fertility (Table 6).

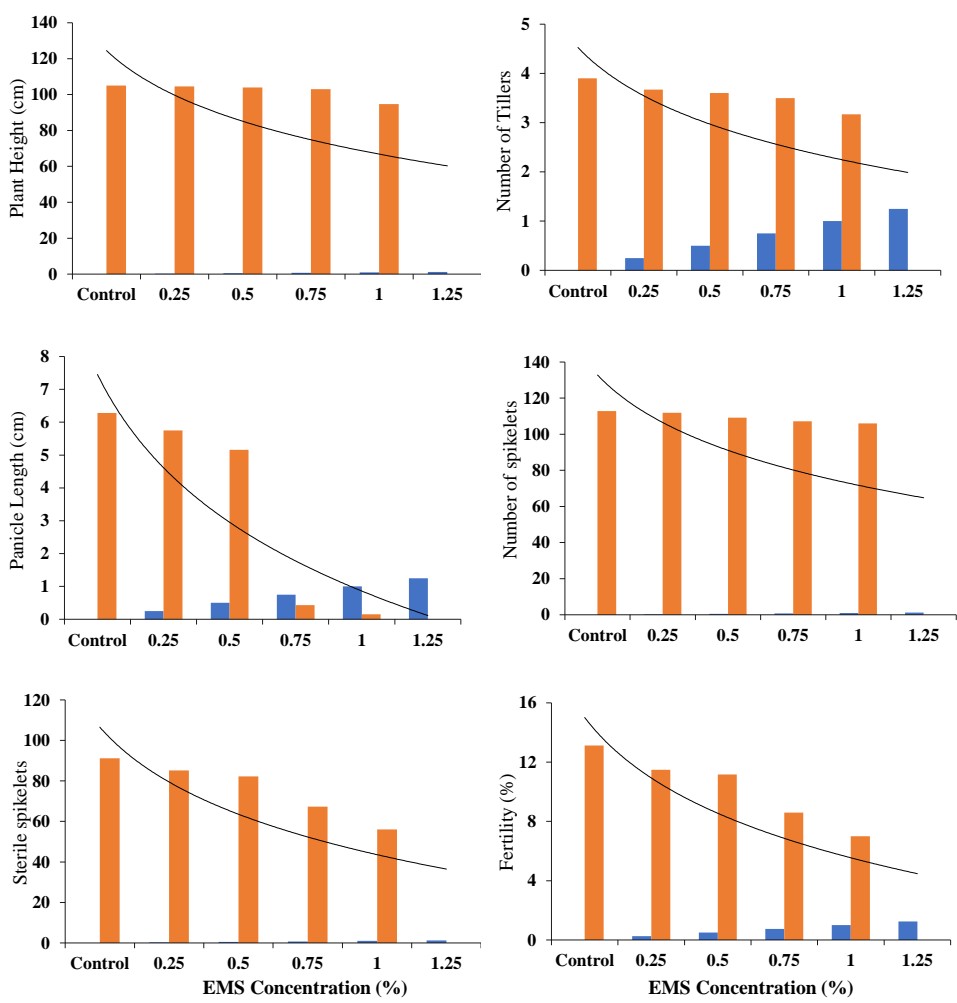

**Figure 5** Effect of different concentrations of EMS mutagenesis on phonological and yield attributes of fragrant rice.

## Lethal dose effects

An EMS treatment at various concentration levels was evaluated on the aromatic rice variety Super Basmati to determine the $LD_{50}$ based on the germination rate, growth, and yield attributes of rice (Fig. 6). The results obtained in this investigation indicated that the germination and all other measured attributes decreased when the EMS dose was increased. The $LD_{50}$ values for seed germination (0.069%), root length (0.6%), shoot length (0.625%), plant height (1.125%), productive tillers (1.125%), panicle length (1.125%), total spikelet (1.126%), sterile spikelet (1.06%), and productivity (1.05%) for Super Basmati rice variety were carefully determined (Fig. 7).

## DISCUSSION

EMS mutagenesis resulted in a major reduction in germination under the prevailing field conditions. As the EMS concentration increased, there was a substantial decrease in seed

**Table 6  Mean value of yield attribute of fragrant rice by following EMS mutagenesis.**

| Treatment | Total spikelet | | Sterile spikelet | | Fertility (%) | |
|---|---|---|---|---|---|---|
| | Observed | % Control | Observed | % Control | Observed | % Control |
| Control | 112.9 | 100 | 91.13 | 100 | 13.12 | 100 |
| 0.25% | 111.9 | 99.11 | 85.2 | 93.49 | 11.48 | 87.5 |
| 0.50% | 109.2 | 96.72 | 82.27 | 90.28 | 11.17 | 85.14 |
| 0.75% | 107.2 | 94.95 | 67.33 | 73.88 | 8.6 | 65.55 |
| 1.00% | 106 | 93.89 | 56 | 61.45 | 7 | 53.35 |
| 1.25% | 0 | 0 | 0 | 0 | 0 | 0 |
| LSD% | 42.23 | | 5.26 | | 0.99 | |
| C.V% | 27.72 | | 4.65 | | 6.50 | |

germination. The EMS has shown to be one of the most potent chemical mutagens as an alkylating agent. According to previous research work, it has been documented that polyploids are more tolerant than diploids (*Chopra, 2005*). The Basmati rice used in the study is also diploid. The percentage reduction in seed germination might have been caused by the influence and impact of mutagens on the meristematic tissues of the seed. The decrease in seed germination at higher dose levels of the mutagens may be attributed to the disturbances at the cellular level with implications at the physiological level.

Earlier research work has shown that in okra (*Abelmoschus esculentus*), germination percentage generally decreased with increasing dose concentrations of gamma rays and the EMS levels (*Gupta et al., 2016*). Reduced germination percentage with increasing doses of gamma radiation has also been reported in pinus (*Ariraman et al., 2014*), rye (*Khah & Verma, 2015*), and chickpea plants (*Shah et al., 2008*). A gradual reduction in germination percentage was also observed with an increase in the concentration of mutagen, reaching more than 50% lethality at 0.5% EMS level in two genotypes of tobacco plants (*Dhakshanamoorthy, Selvaraj & Chidambaram, 2010*). In this study, seeds of the Super Basmati rice variety were treated with chemical mutagen EMS viz., 0.25%, 0.5%, 0.75%, 1%, and 1.25% concentrations. In the laboratory germination test, it was observed that an increase in the level of EMS had an overall adverse effect. Similar results have been reported in a previous study of *capsicum annuum* (*Hasan et al., 2022*) that seeds treated with 1.5% of EMS dose in M1 generations had the lowest germination percentage (−84%) among all treatments. The germination percentage was found to be profoundly inhibited by EMS treatment in two varieties (Co1 and Co2) of soybean plants (*Karthika & Subba, 2006*). The mutagenic reaction is more or less linear with the dosage quantity. Plant survival to maturity is dependent on the type and extent of chromosomal damage according to a previous study on radiation mutation (*Naaz et al., 2022*). Germination inability, plant growth, and survival can be reduced as the occurrence of chromosomal damage increases with growing radiation dose (*Sood et al., 2016*). Furthermore, genes close to the centromere are more sensitive to mutagenic treatment than genes further apart. Significant change in the chlorophyll contents were noticed by the EMS treatment group and increased in the frequency of chlorophyll mutants depends on the dose or concentration of the mutagen

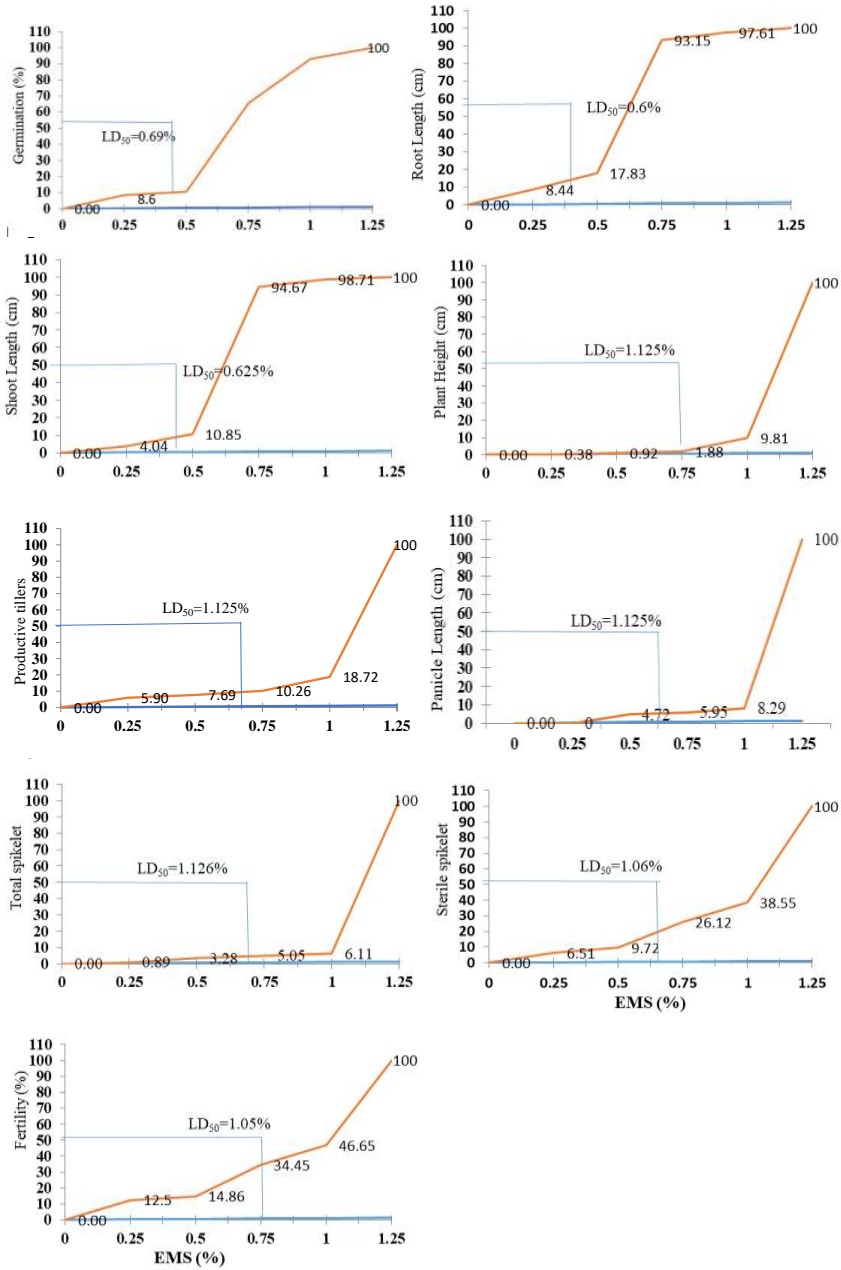

**Figure 6** **Effect of chemical mutagen (EMS) LD50 value on germination, growth, and yield attributes of Super Basmati.**

used (*Bado et al., 2015*). The activation of RNA or protein synthesis may be responsible for the stimulating effect of physical mutation on germination. It can happen after the seeds have been processed during the early stages of germination (*Zhang et al., 2021*). Shoot length is most commonly used as an index to classify and report the biological effects of different physical and chemical mutagens in M1 (*Boyanee, 2015*). Shoot length and the dosage of physical or chemical mutagens have been shown to have a linear relationship.

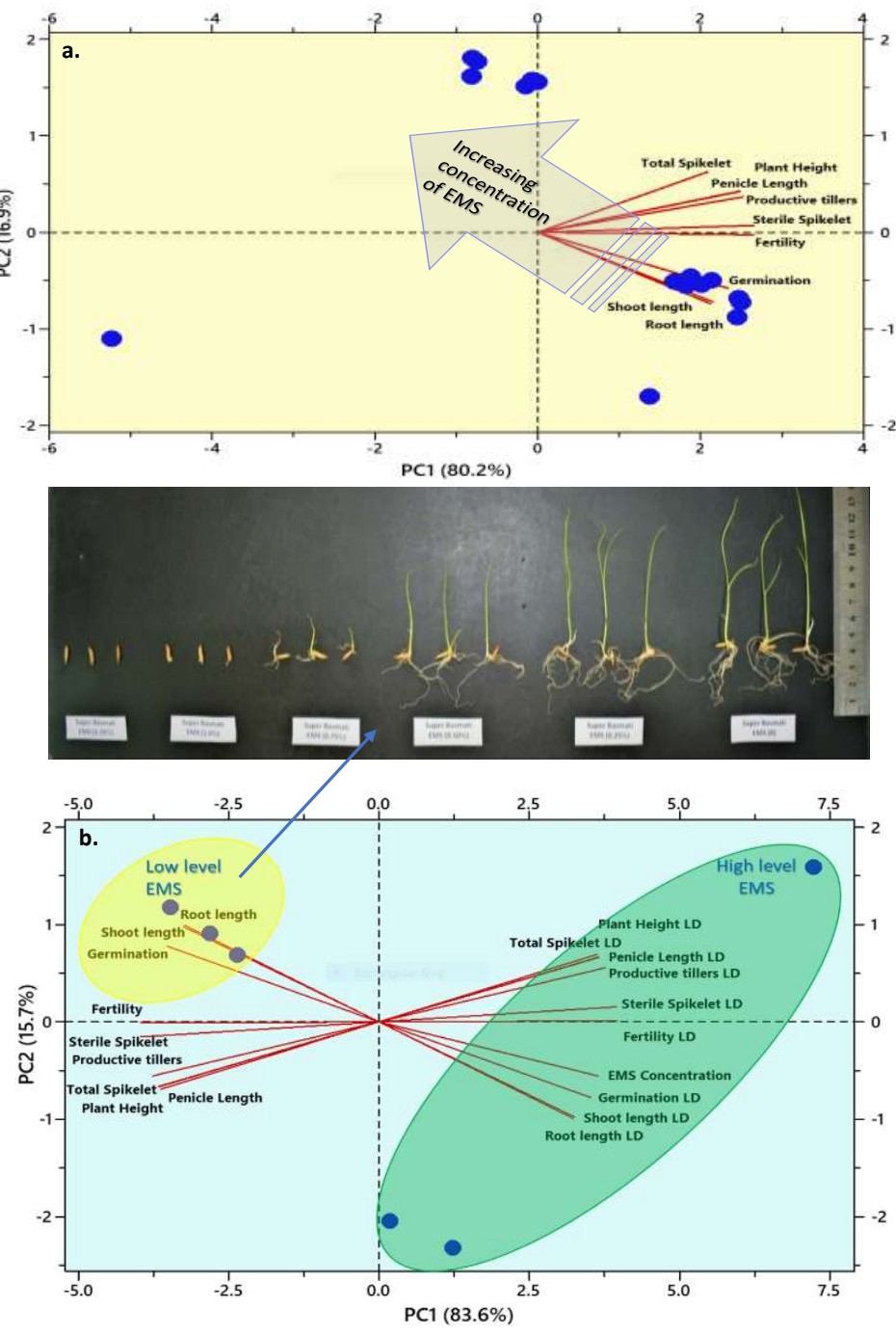

**Figure 7  Biplots from Principal components analysis of the measured parameter of the Super Basmati rice.** Lines indicate determined parameters and circles show the chemical mutagens levels. Image point to changes in the lengths of roots and shoots. The principal components analysis revealed a systematic trend in EMS treated plants. Two groups formed due to mutual association with the determined variables. An arrow indicates that as the concentration of EMS increases, germination, shoot, and root lengths decrease. An outlier (plot a) shown on the negative PC1 and PC2 quadrant represent the highest EMS level at which germination is inhibited. A cluster of measured parameters (plot b) groups traits at low and high levels of mutagen. A low level of EMS dose results in higher fertility and growth parameters such as root length.

Measured data of this study showed that increases in the EMS concentrations caused decreases in shoot length. Our findings also revealed that when the rice variety of Super Basmati was treated with EMS, the shoot length decreased significantly as compared to the control. The concentration of applied EMS had an important impact on the root length of Super Basmati rice. Every subsequent increase in the EMS concentration resulted in a reduction in root length. Enhancement or inhibition of germination, shoot length, and other biological responses are commonly observed in low or high dose treated plants (*Deoli & Hasenstein, 2018*).

According to *Ramchander, Pillai & Ushakumari (2014)*, a low dose of irradiation induces growth stimulation by either modifying the hormonal signaling network in plant cells or growing the cells' anti-oxidative ability. Plants can easily withstand everyday stress factors such as light intensity and temperature variations in the environment. The cell cycle arrests at the G2/M phase during the somatic cell division and various types of damages in the entire genome have been associated with the high dose treatments (*Ahmad et al., 2022*; *Jan, Parween & Siddiqi, 2012*). Variability was assessed in this analysis by the mean values of the shoot and root lengths, both of which decreased as the concentration of EMS increased. When the radiation amount is sufficient to reduce root length percentage, the root lengths do not exceed a few millimeters in size as reported in a physical mutation analysis (*Chaudhuri, 2002*). After irradiation, seeds are unable to germinate due to metabolic disorders (*Sood et al., 2016*).

When rice plants are exposed to lower dosages of mutagens, they exhibit defensive responses that involve structural changes in the photosynthetic machinery. Increasing the concentration of EMS enhanced the concentration of photosynthetic attributes in a linear pattern, however, at 1.25% EMS concentration plant growth and development was affected due to poor rate of germination. Maximum chlorophyll content was seen when *capsicum annuum* was treated with 0.1% EMS for 3 h, according to *Ahmad & Asif (2023)*. *Saba & Mirza (2002)* reported a similar finding by discovering that tomato plants treated with 0.5% EMS for three hours had the maximum chlorophyll content along with other photosynthetic attributes. Enhancement in the water related attributes might be due to the better growth and stay green character of rice plants (*Chaudhari, Verma & Chaudhary, 2015*). However, lower and extremely higher doses could cause a significant change in the DNA of the rice plants, as has been reported for mutagenesis studies in rice (*Abid et al., 2018*; *Viana et al., 2019*). Except for control and 1.25% EMS concentration, all treatments exhibited a notable improvement in water content, but the 1.00% exhibited a decline in the water related attributes. According to *Elyadini et al. (2021)*, one of the two adaption mechanisms maintaining a high level of tissue elasticity or lowering osmotic pressure can contribute to the maintenance of a relatively high value of the relative water content under EMS treatment leading to a positive effect on the rice plants for improving its overall productivity.

The enzymatic antioxidants are essential enzymes in plant cells that remove $H_2O_2$ from various organelles like the cytosol and chloroplast to prevent oxidative damage (*Zahra et al., 2021*). High APX activity was observed in this study and various researchers have also observed an increase in APX activity during the increased concentrations of EMS

treatments (*Hamid et al., 2015*). Therefore, the production of reactive oxygen species in cells that lack water content causes cell damage, which eventually culminates in cell death (*Bali & Sidhu, 2019*). The balance in the enzymatic antioxidants, whose activity was raised at moderate levels of EMS treatments, is one of the antioxidant systems that regulate oxidative stress through a variety of adaptive ways (*Palace et al., 1998*). Under higher levels of EMS, enzymatic antioxidant activity was found to be elevated in wheat, as it was in many other crop species (*Devi, Kaur & Gupta, 2012*). Cell walls, vacuoles, extracellular spaces, and cytosol, all contain APX. This enzyme, which is known as a stress indicator, has a broad range of phenolic substrate selectivity and is attracted to $H_2O_2$ than catalase. It can use $H_2O_2$ to produce phenoxy chemicals, which ultimately polymerize lignin, a component of the cell wall (*Štolfa et al., 2015*).

The EMS-treated seeds may develop a mutant basmati rice variety. This is possibly due to the pleiotropic impact of mutated genes or mutations on various genome loci (*Muqaddasi & Arif, 2012*). Several morphological mutations in legume plants have also been identified (*Goyal et al., 2019*) and few of mutations have been shown to affect multiple attributes. A combination of the elevated amount of dose and the period of treatment resulted in higher seedling death and lower yield in the plant characteristics in the EMS-treated seeds. Similar results were recorded in an experiment of EMS-treated fenugreek seeds, where no callus cultures were developed when treated with EMS levels above 1% (*Basu, Acharya & Thomas, 2008*). In this analysis, $LD_{50}$ values for yield contributing traits included the plant height (1.125%), productive tillers (1.125%), panicle length (1.125%), complete spikelet (1.126%), sterile spikelet (1.06%), and fertility (1.05%) which were found in seeds treated with 0, 0.25, 0.5, 0.75, 1, and 1.25 percent EMS, resulting in an inverse association between all of these yielding traits (*Kozgar, Goyal & Khan, 2011*). The efficacy of the current study decreased as the concentration of EMS increased. This observation was also confirmed by the findings in black gram (*Usharani & Kumar, 2015*), chickpea (*Singh et al., 2015*) and cowpea plants (*Nair & Mehta, 2014*).

The variation in $LD_{50}$ for the Super Basmati rice variety at the different EMS (%) concentrations has been observed in mutation studies, and it is thought to be mainly due to the biological material, scale, maturity, hardness, and moisture content at the time of exposure of breeding material (*Thakur, Paul & Kumar, 2020*). There is sufficient evidence that the radiation-induced sterility of M1 panicles is passed on to subsequent generations (*Jyothilekshmi, 2012*). Physiological damage induces a significant portion of sterility, which is not passed on to the next generation. It is found in this research work that with the increasing doses of mutagen treatments, induced panicle sterility. These findings are consistent with those of previous researchers (*El-Degwy, 2013*; *Siddiqui & Singh, 2010*) who found that gamma-ray treatment caused rice plants to become highly sterile. In determining the yield potential of these mutants, it will be vital to analyze the heritability in a multi-location yield trial that incorporates suitable experimental design to assess whether these mutations will perform consistently across different environments.

## CONCLUSIONS

Physicochemical mutagenesis has been employed to produce genetic variability in crops including rice plants. The EMS induced mutagenesis is a promising exploratory tool to search for novel players for improving agronomic and yield contributing traits. Germination, seedling growth, and yield attributes were significantly influenced by variations in EMS concentration treatments. There was no germination observed upon the application of a 1.25% concentration of EMS treatment for seed germination and 50% germination was recorded between EMS 0.50% and EMS 0.75% treatments. After the cultivation of rice plants of the M1 generation in the field, it was observed that $LD_{50}$ for fertility occurred at EMS 1.0% for the investigated rice variety. The EMS treatment demonstrated a negative biological influence such as reduced germination and abnormal seedling development of Basmati rice plants. It is, therefore, concluded that for creating genetic variability in the rice variety of Super Basmati, the EMS doses from 0.5% to 0.75% are more useful and effective for improving the overall performance of fragrant rice. Furthermore, mutants with yield related value-added traits will be available for the scientific community for advanced level research and will also serve as a public genetic resource for development and breeding programs.

### Funding

This work was supported by the Researchers Supporting Project number (RSP2023R194), King Saud University, Riyadh, Saudi Arabia. The funders had no role in study design, data collection and analysis, decision to publish, or preparation of the manuscript.

### Grant Disclosures

The following grant information was disclosed by the authors:
King Saud University, Riyadh, Saudi Arabia: RSP2023R194.

### Competing Interests

The authors declare there are no competing interests.

### Author Contributions

- Areeqa Shamshad conceived and designed the experiments, performed the experiments, authored or reviewed drafts of the article, and approved the final draft.
- Muhammad Rashid conceived and designed the experiments, performed the experiments, authored or reviewed drafts of the article, and approved the final draft.
- Ljupcho Jankuloski analyzed the data, authored or reviewed drafts of the article, and approved the final draft.
- Kamran Ashraf analyzed the data, prepared figures and/or tables, and approved the final draft.
- Khawar Sultan analyzed the data, prepared figures and/or tables, and approved the final draft.

- Saud Alamri conceived and designed the experiments, authored or reviewed drafts of the article, and approved the final draft.
- Manzer H. Siddiqui conceived and designed the experiments, authored or reviewed drafts of the article, and approved the final draft.
- Tehzeem Munir analyzed the data, prepared figures and/or tables, and approved the final draft.
- Qamar uz Zaman conceived and designed the experiments, authored or reviewed drafts of the article, and approved the final draft.

## Data Availability

The raw measurements are available in the Supplemental Files.

## Supplemental Information

Supplemental information for this article can be found online at http://dx.doi.org/10.7717/peerj.15821#supplemental-information.

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
