# Peer review of "Effect of ethyl methanesulfonate mediated mutation for enhancing morpho-physio-biochemical and yield contributing traits of fragrant rice"

_PeerJ, doi:10.7717/peerj.15821_

## Round 0.1 · original submission · Major Revisions

Dear authors,

Thank you for submitting your work to PeerJ. Two experts in the field have addressed your manuscript. While one of the reviewers is indicating simply that you should revise the reference list, reviewer 2 and I consider that the manuscript requires much work to move forward. I kindly request that you provide a thoroughly revised version of your manuscript.

Reviewer 2 identified key issues in the manuscript that need attention and must be addressed in totality. I think that the manuscript lacks details of the experimental approach and raw data regarding the phenotype of the mutagenized plants. Also, Table 1 and 2, replace "Actual" with "observed". In Figures 2, 3, and 5 error bars must be included as well as the statistical values for each set.

English must be revised throughout the manuscript, there are phrases with missing articles or incorrect punctuation.

I hope these comments are useful for the authors.

Kind regards,
Bernardo

·

Basic reporting

no comment

Experimental design

no comment

Validity of the findings

no comment

Additional comments

Please check the cited references to ensure they are accurate.

·

Basic reporting

The authors present an interesting and well-developed work, from the experimental point of view. However, it is necessary to make important changes in the structure of the introduction, including additional information, which allows the reader to highlight the importance of the research work.
It is important to improve the quality of the figures so that the reader can appreciate the results obtained in detail and can relate them from the text.
Finally, in the discussion is important to include their perspectives about the possible genetic affectations that control the particular characteristics of the Basmati rice variety and what they would expect in the following generations of plants, after treatment with EMS.
A previous work “Ethyle methane sulphonate (EMS) induced mutagenic attempts to create genetic variability in Basmati rice (Watoo et al., 2012)”, already reported the effect of EMS on germination and some agronomic traits of two Basmati rice varieties, which are not mentioned in this work. This submitted paper presents some additional aspects evaluated in plants treated with EMS, but it is essential that authors analyze and compare the common results and justify the relevance of the data that may be new, depending on the dose, exposure time and number of individuals analyzed.
At this time, I cannot comment or suggest detailed changes to the document because, in my humble opinion, the document requires substantial changes.

Experimental design

I suggest that the methods be written so that there is no doubt about the protocols that were followed in the experiments. EMS treatment is an essential part of the work and it is not clear from the methods how long the seeds were exposed to the mutagenic agent. It is also not clear how many individual samples, from each treatment, were used to analyze the different parameters selected (chlorophyll content, water, enzymatic activities).

Validity of the findings

I could not evaluate this aspect because I do not know the number of individuals analyzed. From what I could observe in the document, the data does not have the requirements to be robust, solid and controlled.
One of their main conclusions is te same reported by Watoo et al (2012) "The efficiency of EMS was higher at lower concentration. Results suggest that using a dose 0.5 to 1.0% of EMS for 6 h can induce mutations in rice"

Additional comments

I list some of my observations to the manuscript:
-I understand that Basmati is a particular variety of rice and it would be interesting to know the level of consumption of this variety compared to the more cultivated ones. Is anything known about the origin of the variety? Is the size of its genome known, the ratio of G/C bases, which would be the target of the alkylating agent? Additionally, it would be necessary to briefly include in the introduction something that helps to justify the parameters that are going to be analyzed in the treated plants. Like the chlorophyll content, the water content and the enzymatic activities, which are related to the stress response. In part, this explanation is included at the end, in the discussion, but I consider that it could be presented from the introduction, facilitating the justification and understanding of the experimental work you did.
https://www.frontiersin.org/articles/10.3389/fgene.2020.00086/full this paper mentions some important genetic characteristics for Basmati rice.

- Line 62 “cause a broad range of functional mutations in rice (Cho et al., 2010) “ Could you clarify in what kind of rice varieties the other authors have done the mutagenesis?
- Line 66-67 alkylating agents that have diverse effects on DNA… Could you very briefly point out the type of effects that are caused in DNA?
- Line 80-81 New ventures have been launched in recent years to produce EMS-induced rice varieties in research institutes. The first reference corresponds to Arabidopsis and the second gives an example of a single variety of rice. Can you provide some other examples?
- Line 81-84The LD50 dose 
is first calculated and then used to determine the best dose for inducing mutations (Blakely et al., 1998; Bhattramakki et al., 2002; Borevitz et al 2003). I cannot find in the references the relationship with the doses of EMS to generate mutations, each reference deals with different aspects, the first is about radiobiology, the second is about SNPs identified in natural populations of maize and the third is about the comparison of SNPs in two Arabidopsis ecotypes. Finally, this reference Baghery et al., 2016 talks about the doses for a Malvaceae plant, and it could be clarified in the text that it is a totally different species from rice.
- Line 85-86 Basmati rice has a narrow genetic base and broadening its genetic base using non-basmati rice material may affect its quality attributes. Quality attributes will also be affected by an EMS treatment. When making crosses with other varieties, hybrids with characteristics that are totally removed from the characteristics of the variety of interest will be generated. Transgenic approaches have problems with GMOs. I don't understand what you mean by this sentence. There are many positions about the generation and use of transgenics but I think that it can be better justified why this possibility would not be viable in the case of the variety of rice with which you work, and include comments about a CRISPR edition. In each case it is required to know and implement transformation methods and I do not know if in the case of Basmanti rice have been implemented. In addition, this would give them more specific modifications, not a degree of genetic variability. I think these aspects should be considered and clarified in your text.
- Line 89-90 …creating genetic variability without compromising the quality attributes. …creating genetic variability maintaining the particular characteristics that make the Basmati variety special, without compromising the quality attributes.
- Line 94 requires a reference
- Is it possible that EMS treatment causes a mutation in some of the relevant genes that control fragrant traits in Basmati traits? Could you evaluate this possibility in the plants treated with EMS in M1?
- Super Basmati seeds were soaked in ~100 mL ultrapure water (Sigma, grade) up to a height of 5 cm above the seeds and stored at room temperature overnight for 20 hours. EMS (v/v) concentrations of 0.25, 0.50, 0.75, 1.0, and 1.25 % were then added to 50 mL of the decanted water. The seeds were then transferred and rinsed with ultrapure water (~100 mL, 5 times, & 4 minutes each) and 200 mL ultrapure water after being incubated at room temperature for 12 hours (4 times, and 15 minutes each). The seeds thus processed were then washed in continuously running tap water for about 4 hours before being placed in Petri dishes for further analysis (Fig. 1). The 400 seeds were distributed in 6 beakers? as seen in the figure. Was the water where the seeds incubated absorbed by them? If there was water left, was it removed and 50mL of water containing the different concentrations of EMS added? They were washed with ultrapure water five times for 4 minutes each time. 200mL of ultrapure water were added and incubated at room temperature (at what approximate temperature?), for 12 hours. It is not understood why it says 4 times and 15 minutes each if they have already indicated that it is a 12-hour incubation. Figure 1 has more photos that do not correspond to the treatment of the seeds (maybe only the photos with letters a, b and c. I consider that it is not necessary to include a photograph of a person doing any of the washings. None of the texts in the figures can be read, where the treatments are indicated, when zooming in on the figures they look very blurry. I suggest that the original photographs be included, with legible treatment indications.
- How many seeds were used for each treatment. After the treatment with EMS how the seeds were treated to analyze % germination. How many of them were planted in the field or soil; were sown or transplanted as seedlings. The analyzed plant material was collected from those plants grown in the field, all at the same stage of development (tillering) or all at the same age after sowing or transplanting.
- Line 119. Sample of plant’s leaf was crushed with ratio of 5g take in test tubes having 85%... the sentence is not clear enough. It could be clarified, by writing differently Samples (5 g) of leaves from each treatment were collected on test tubes containing acetone 85%; each sample was macerated in the same tube. Samples were kept in the dark for 24 hours to allow the extraction of photosynthetic pigments. Tubes were centrifuged 10 min at 4000 x g at 4oC to remove cellular debris. Supernatants were measured in a spectrophotometer (Halo …… ) to determine chlorophyll contents.
- Line 124. Is freezing and thawing enough to obtain the sap contained in the tissue or is an additional step necessary? Are the samples frozen and thawed in a tube, are the tissue removed, or are tubes with tissue centrifuged?
- Line 161-163. In the case of seed germination, data indicate (Fig. 2) that all of the EMS treatments showed an inhibitory effect by suppressing and restraining growth as compared to the control. This sentence gives the impression that all doses of EMS had the same effect. It can be written in another way, so as to clarify that in all cases there is an effect, but that it occurs to different degrees depending on the dose. (In the case of seed germination, data indicate that EMS had an effect retarding growth, or even inhibiting it, depending on the dose applied, as compared to the control).

- Line 232-233. According to previous research, it has been documented that polyploids are more tolerant than diploids. Is the Basmati variety diploid?
- Line 235-236 The decrease in seed germination at higher dose levels of the mutagens may be attributed to the disturbances at the cellular level (caused either at the physiological level). 
…. with implications at the physiological level.
- Line 237. Earlier research work has shown that in okra (Abelmoschus esculentus) germination. Cursive for scientific names 

- Line 287 and 289. you mention examples of the effect of EMS for three hours. In the methods section, you do not indicate how long the EMS was in contact with the rice seeds
- Line 292. .. lower and extreme higher doses cause a significant change in the DNA of the rice plants. (.. could cause a significant change in the DNA of the rice plants, as has been reported for …) could you provide some examples.

I hope that these suggestions and comments are useful for the authors in improving the presentation of their results. This is another paper that could be useful in restructuring your manuscript. Mutagenesis in Rice: The Basis for Breeding a New Super Plant (Vívian Ebeling Viana et al., 2019)

---

## Round 0.2 · Minor Revisions

Dear authors,
First of all, I apologize for the delay on the revisions. Secondly, the expert reviewer have addressed your corrections and have identified numerous issues that need attending. However, the reviewer suggests acceptance of the manuscript but the authors should commit to provide a thorough revision of the issues the manuscript still has. I concur with the decision of the reviewer, the manuscript is relevant but still need some work, specially on the pertinence of some of the references used, the writing in the parts the reviewer indicates and some other details that should be corrected before publications. Nevertheless, congratulations for your work and thank you for submitting this manuscript to PeerJ.
I thank you for your patience.
All the best for all of you and for your research,
Bernardo

·

Basic reporting

this is the second review and all my comments are flagged in the pdf file i am sharing

Experimental design

this is the second review and all my comments are flagged in the pdf file i am sharing

Validity of the findings

this is the second review and all my comments are flagged in the pdf file i am sharing

Additional comments

The provided PDF file is difficult to understand. The document seems to start multiple times and it is not possible to track changes based on line number. For this reason, I based my review on the word file, to which the authors made corrections. This new document has additional information in the introduction and the authors have addressed almost all of the suggestions that were made in the first revision. However, there are still details to attend to. All new suggestions and comments are indicated in the document that I am sharing.
In my opinion, it is still important to consider the closest works that have already been published and that make use of the same mutagenic agent in the same plant system.
I consider that its publication can be approved as long as editing aspects are corrected and the comments that are indicated in the PDF file are considered

---

## Round 0.3 · accepted · Accept

Dear authors,

Thank you so much for correcting the remaining issues. The manuscript is now suitable for publication.

Best regards,
Bernardo